# Study of Helium Swelling in Nitride Ceramics at Different Irradiation Temperatures

**DOI:** 10.3390/ma12152415

**Published:** 2019-07-29

**Authors:** Maxim. V. Zdorovets, Kanat Dukenbayev, Artem. L. Kozlovskiy

**Affiliations:** 1Engineering Profile Laboratory, L.N. Gumilyov Eurasian National University, Astana 010008, Kazakhstan; 2Laboratory of Solid State Physics, The Institute of Nuclear Physics, Almaty 050032, Kazakhstan; 3Department of Intelligent Information Technologies, Ural Federal University, 620075 Yekaterinburg, Russia; 4School of Engineering, Nazarbayev University, Nur-Sultan 010008, Kazakhstan; 5Laboratory of Additive Technologies, Kazakh-Russian International University, Aktobe 030006, Kazakhstan

**Keywords:** swelling, helium bubbling, nitride ceramics, construction materials, destruction

## Abstract

This paper presents the results of a systematic study of helium swelling and the subsequent process of degradation of the near-surface layer of aluminum-based nitride ceramics. The samples were irradiated with 40 keV He^2+^ ions at temperatures of 300 and 1000 K with a fluence of 1 × 10^17^–5 × 10^17^ ions/cm^2^. The choice of radiation doses and temperature conditions was due to the possibility of simulating reactor tests of structural materials. It has been established that an increase in the irradiation fluence leads to the formation of large agglomerates of clusters of helium bubbles, as well as an increase in the degree of roughness and waviness of the surface with the formation of crater-like inclusions. In the case of irradiation at high temperatures, there was a slight decrease in the average size of helium inclusions compared with irradiation at room temperature. However, the density of inclusions and surface roughness were much higher. It is established that irradiation at room temperatures leads to a sharp decrease in ceramics density, as well as deformation of the crystal structure due to an increase in the density of dislocations and macrostresses in the structure. The decrease in ceramics density due to the formation of helium inclusions led to an increase in porosity and a defective fraction in the structure of the surface layer of ceramics.

## 1. Introduction

One of the most acute problems of structural materials used in the nuclear industry is the accumulation of helium and the further helium embrittlement of material surface layer [1,2,3]. Degradation of the surface can lead to a decrease in thermal conductivity, deterioration of insulation and optical characteristics, which leads to destabilization of reactor operation. In this case, the accumulation of helium in the structure of structural materials occurs as a result of its low solubility, as well as a high rate of mobility and the possibility of agglomeration. The emergence of helium in the structure occurs as a result of the initiated nuclear reactions under the effect of neutrons, and in the case of surface layer contact with the coolant [4,5,6,7,8,9,10]. Moreover, in the case of nuclear reactions, helium occurs in the entire volume of the material—that somewhat reduces the probability of agglomeration in the surface layer, while in the case of contact with the coolant, the main accumulation occurs in the small surface layer 0.5–1.0 μm thick. The accumulation of helium in the surface layer led to swelling, flaking, and partial destruction of the surface of structural materials, which has a negative impact on the performance of materials [11,12,13]. The swelling and degradation processes occur as a result of the agglomeration of helium and the subsequent formation in the structure of gas inclusions, the average size of which can vary from 100 nm to 1 μm [13,14,15,16]. The appearance of gas inclusions and their further evolution limits the scope and lifetime of many structural materials in which a low content of helium can lead to disastrous consequences [17,18,19,20,21].

In recent years, more and more research has been devoted to the study of radiation resistance and defect formation processes in ceramic materials based on nitrides or carbides [22,23,24,25,26,27,28]. The interest in these materials is due to the possibility of their use as structural materials of high-temperature nuclear reactors, due to the high melting temperature (above 2000–2500 °C), insulation characteristics, resistance to aggressive media, etc. [25,26,27,28,29,30]. One of the most promising materials with all these characteristics is aluminum nitride [31,32,33,34,35]. As a rule, aluminum nitride-based ceramics are obtained by sintering oxide powders in a nitrogen- or carbon-containing medium at high temperatures and pressures. At the same time, the sintering temperature ranges from 660 to 2000 K, the choice of which is due to the chemical reactions of synthesis [36,37]. Therefore, in the researches [36,37] a detailed description of methods for producing aluminum nitride by various methods obtained in different laboratory conditions was presented. In most cases, the sintering of nitride ceramics is accompanied by the release of gases CO, N_2_, H_2_. Furthermore, the reaction products in the synthesis process, depending on the gas medium, can be oxide, oxonitride, carbide inclusions, and gas-filled pores. In most cases, industrial production of aluminum nitride is carried out using the method of carbothermic reduction of powders Al_2_O_3_ and C (s) in the medium N_2_ (g) at a temperature of about 2000 K at a pressure p = 0.12–0.26 atm. As a result of annealing, AlN-type ceramics and CO (g) gas were obtained, with a small part of Al_2_O_3_ being equally likely to dissolve in the ceramic structure. At the same time, the appearance of impurity inclusions can lead to additional distortions and deformations of the crystal structure, as well as changes in the phase composition of ceramics. In this case, the use of aluminum oxide in the technological process of production leads to the presence of small impurity inclusions of oxide phases in the structure, which can have a significant impact on the radiation resistance of AlN ceramics. Previously, our research team conducted a series of studies of the radiation resistance of AlN ceramics to the effects of irradiation with heavy ions with energies close to the energies of fission fragments of uranium nuclei [38,39,40,41], as well as the radiation degradation of structural and mechanical properties as a result of carbon accumulation in the surface layer [42,43,44,45]. In the course of these studies, we tested the method of testing and evaluating radiation resistance using atomic force microscopy, X-ray diffraction, scanning electron microscopy, and optical methods of analysis [40,41,42,43,44]. In this study, the results of studying the radiation resistance to helium swelling under irradiation conditions close to reactor tests are presented.

## 2. Materials and Methods

The samples under study were polycrystalline structures of the hexagonal type of aluminum nitride, similar to the structure of wurtzite. The original ceramic samples were purchased from Mingrui Ceramic Tech. Co. Ltd. (Changan Town, Guangdong, China) specializing in the production of various ceramics for industrial applications and research. The thickness of the samples was 10 μm. Before irradiation, the samples were polished to achieve a surface roughness of no more than 3–4 nm. Figure 1 shows the AFM image of the surface of initial sample and the X-ray diffraction pattern.

As can be seen from the presented data, the initial sample was a polycrystalline structure with selected textural directions (100), (002), and (101), characteristic of hexagonal type of structures. The crystallite size for initial sample was not more than 95–100 nm. The lattice parameters were a = 3.0957 Å, c = 4.9557 Å, V = 41.13 Å^3^. The presence of low-intensity peaks characteristic of the Al_2_O_3_ phase was due to the processes of technology for producing nitride ceramics by sintering. Figure 1c shows a SEM image of the surface of original ceramics before irradiation. As can be seen from the presented data, the surface of the initial sample is characterized by relative smoothness, without roughness and visible defects.

Helium swelling was simulated by irradiating nitride ceramics with low-energy He^2+^ ions with an energy of 40 keV at temperatures of 300 and 1000 K with a fluence of 1 × 10^17^–5 × 10^17^ ions/cm^2^. Irradiation was carried out at the DC-60 heavy ion accelerator (Nursultan, Kazakhstan). The ion path lengths were calculated using the Kinchin-Pease model using the SRIM Pro 2013 program and 240 ± 10 nm, the radial deviation is 60 ± 5 nm, the energy loss of ions on electrons is dE/dx_elec_ = 0.184 keV/nm, the energy losses on nuclei are dE/dx_nuclear_ = 0.07 keV/nm [45].

The study of changes in structural characteristics and morphology of samples was carried out using atomic force microscopy (AIST-NT SPM microscope, AIST-NT Inc., Novato, CA, USA, Scan mode was the following: AC-Mode (Non-Contact Mode), scan rate/scan frequency is 0.4 Hz, resolution XY 700 × 700, scan size XY 15 × 15 um, Z is automatic), X-ray diffraction (XRD D8 ADVANCE ECO diffractometer (Bruker, Karlsruhe, Germany) using CuKα radiation, identify the phases, and study the crystal structure, the software Bruker AXSDIFFRAC.EVAv.4.2, and the international ICDD PDF-2 database), and scanning electron microscopy (SEM, Hitachi TM3030 (Hitachi Ltd., Chiyoda, Tokyo, Japan)). A detailed description of the methods and instrumentation base was presented in our researches [38,39,40,41,42,43,44].

The analysis of structural changes was carried out using the following calculation formulas. Griffith’s criterion characterizing the changes in the strength characteristics and crack resistance of materials as a result of external influences was calculated by the Equation (1):(1)$$K_{cr}=S^{2}L$$
where *S* is the average stress, *L* is the crystallite size. The calculation of average stress in the crystal structure was carried out using the following Equation (2):(2)$$S=\varepsilon \cdot \frac{Y}{2G}$$
where ε is the structure deformation coefficient, *Y* is the Young’s modulus, and *G* is the Poisson’s ratio for the material.

The dislocation density in the structure carrying information about the defective component in the crystal lattice was calculated according to the Equation (3):(3)$$\delta =\frac{1}{L^{2}}$$
where *L* is the size of crystallites. The average crystallite size by the Scherer Equation (4):(4)$$L=\frac{k\lambda }{\beta \cos\theta }$$
where *k* = 0.9 is the dimensionless particle shape factor (Scherer constant), *λ* = 1.54Å is the wavelength of the X-ray radiation, *β* is the half-width of the reflex at half-height (FWHM), and *θ* is the diffraction angle (Bragg angle).

The value of the displacements was estimated by measuring the ratios of the two most intense lines of the same sample before and after irradiation and calculated by the Equation (5):(5)U2=3a2ln[(I1I2)irradiated/(I1I2)initial]/4π2[(h22+k22+l22)−(h12+k12+l12)]
where *a* is the lattice parameter, (*I*_1_/*I*_2_)*_initial_*, (*I*_1_/*I*_2_)*_irradiated_* is the ratio of the intensities of the diffraction lines before and after irradiation, respectively.

The density of the material was calculated using the Equation (6):(6)$$p=\frac{1.6602\sum AZ}{V_{o}}$$
where *V*_0_ is the volume of the unit cell, *Z* is the number of atoms in the crystal cell, *A* is the atomic weight of atoms. The concentration of amorphous inclusions in the crystal structure of the investigated nanotubes was according to the Equation (7):(7)Pdil=(1−pp0)×100%
where *p*_0_ is the density of the reference sample.

## 3. Results and Discussions

Figure 2 shows the dynamics of changes in the surface morphology of ceramics as a result of irradiation at different temperatures performed using the AFM method. As can be seen from the presented data, an increase in the dose of radiation above 1.0 × 10^17^ ion/cm^2^ at an irradiation temperature of 300 K in the surface layer, the formation of spherical inclusions whose average size varies from 300 to 400 nm, was observed. In this case, the largest accumulation was observed near the grain boundaries, which served as the accumulation of defects. A further increase in the fluence of radiation led to the formation of large agglomerates of clusters, as well as an increase in the degree of roughness and waviness of the surface with the formation of crater-like inclusions.

In the case of irradiation at a temperature of 1000 K, the degradation of the surface layer was different. Increase in irradiation dose led to a large formation of crater-like inclusions, which indicated a partial sputtering of the surface as a result of irradiation. In this case, in the case of irradiation at high temperatures, no pronounced spherical inclusions were observed on ceramics surface. Figure 3 presents the results of assessing the dynamics of changes in the size of inclusions formed as a result of irradiation.

As can be seen from the presented data, an increase in the irradiation fluence led not only to an increase in the density of inclusions formed, but also to their enlargement and agglomeration. Moreover, in the case of irradiation at a temperature of 300 K with a maximum fluence (5.0 × 10^17^ ions/cm^2^), the change in the size of inclusions was described by two maxima of the Gaussian distribution, which indicated the presence of different generations of helium inclusions. The appearance of the second generation occurred as a result of the partial destruction of the first generation of helium bubbles as a result of an increase in the concentration of internal stresses inside the spheres, which led to their destruction with subsequent processes of degradation and peeling of the surface. Furthermore, new helium bubbles of smaller size were formed in places of defects accumulation. In the case of high irradiation temperatures, the migration rate of helium was accelerated due to the increase in thermal vibrations of atoms in the lattice, as well as the partial annihilation of radiation point defects as a result of thermal exposure. As a result, helium agglomeration in the structure of the surface layer occurred less intensively due to a decrease in the concentration of point defects and partial recombination of defects in the structure. In this case, in the case of irradiation at high temperatures, there was a slight decrease in the average size of helium inclusions in comparison with irradiation at room temperature, however, the density of inclusions and surface roughness were much higher. A change in the surface relief during irradiation at a temperature of 1000 K might be due to the accumulation of only helium ions, unlike irradiation at a temperature of 300 K at which the swelling processes were affected by both helium accumulation and vacancy swelling. In this case, an increase in the vacancy density occurred due to defects accumulation as a result of irradiation, while at high temperatures most of vacancies and point defects annihilated and recombined due to an increase in the thermal vibrations of atoms. Figure 4 shows SEM images of ceramics surface after irradiation.

As can be seen from the presented data, in the case of an increase in the irradiation temperature from 300 K to 1000 K, an increase in the formation of microcracks and grain boundaries in the surface layer was observed. The presence of a large number of helium inclusions in the surface layer of ceramics irradiated at a temperature of 300 K could lead to a sharp increase in porosity and partial swelling, which led to a deterioration of strength characteristics of the material and reduce the lifetime. Figure 5 shows the dynamics of the dependence of change in hardness of the surface layer performed using the indentation method.

For samples irradiated at room temperature, the greatest change in hardness and strength was observed in the surface layer with a thickness of 200–300 nm, while the increase in fluence affected not only the decrease in hardness of the surface layer, but also a slight increase in depth of the deformed layer. This change might be due to processes of helium migration and subsequent accumulation not only in the surface layer, but also partial migration into the material, as well as the formation of cascades of secondary defects that migrated to a depth exceeding the mean free path of helium ions in aluminum nitride. For samples irradiated at a temperature of 1000 K, a smaller change in the hardness of the surface layer was observed, however, a significant increase in depth of the defect region exceeding 350–400 nm for the maximum dose of irradiation. This increase in depth of the defect zone might be due to the high mobility of helium ions in the structure, as well as accelerated migration into the material. However, a slight increase in the hardness of irradiated materials at high temperatures compared with samples irradiated at room temperatures could be due to partial recombination of defects at high temperatures, which reduced the concentration of the vacancy and dislocation densities of defects in the structure.

Figure 6 shows the dynamics of changes in the magnitude of crack formation, estimated using the Griffith criterion, as well as the dynamics of changes in the dislocation density of defects in the structure of ceramics as a result of irradiation.

According to the data obtained, an increase in the irradiation fluence led to a decrease in the Griffith criterion, which indicated a decrease in the crack resistance of the material of ceramics due to partial degradation of the surface layer and helium swelling. In this case, the largest change was observed for samples in which the presence of helium inclusions of the first and second generations was observed. Furthermore, as a result of an increase in the irradiation fluence, an increase in the dislocation density of defects in the structure of ceramics was observed. However, for samples irradiated at a temperature of 1000 K, a slight decrease in the dislocation density was observed, which was caused by partial recombination of defects in the structure as a result of thermal annealing of defects. As a rule, the dislocation density in ceramic materials in the initial state was 10^14^–10^16^ cm^−2^; such a value of dislocation density was due to the small size of grains or crystallites, as well as the presence of various impurities affecting the dislocation density and its distribution near the grain boundaries. At the same time, an increase in the dislocation density in the structure above 10^17^ cm^−2^ led to a sharp degradation of ceramics, due to the partial destruction of crystal and chemical bonds, as well as the fragmentation of crystallites as a result of external influences. According to the data obtained, the dislocation density in the structure of ceramics did not exceed 10^15^ cm^−2^, which confirmed the data on the high strength of ceramics after irradiation.

One of the most important indicators of changes in structural parameters as a result of irradiation and helium swelling was the change in the diffraction pattern of the samples under investigation as a result of irradiation. Figure 7a,c shows the dynamics of changes in the main diffraction maxima as a result of irradiation at different temperatures. The choice of the range for estimating the change was due to the polycrystalline hexagonal structure of ceramics. It was characterized by three main diffraction reflexes corresponding to the three textural directions.

As can be seen from the presented data, for the samples irradiated at a temperature of 300 K at a fluence of irradiation of 1.0 × 10^17^ ion/cm^2^ there was a slight change in the intensity of the diffraction maxima, as well as the broadening of the diffraction lines. A slight displacement of the maxima in the region of small angles indicated a change in interplanar distances and crystal lattice parameters as a result of irradiation. An increase in the irradiation fluence led to a decrease in intensity and an increase in asymmetry of the diffraction maxima, which indicated an increase in distortion and deformation in the structure as a result of irradiation. The formation of helium inclusions in the structure leads to a partial disorientation of texture directions, as well as a change in the size of crystallites, as evidenced by the change in dislocation density. The broadening of the diffraction peaks can be caused both by the size effect and by distorting factors that arise as a result of irradiation. The change in distorting factors in the structure was due to the accumulation of helium with the subsequent formation of helium inclusions, leading to the deformation and degradation of the crystal lattice. Furthermore, for samples irradiated at high temperatures with a maximum irradiation fluence, no additional maxima appeared. Figure 8a,b shows the comparative dynamics of the main diffraction maximum (100) depending on the fluence and the irradiation temperature. As can be seen from the presented data at the irradiation temperature of 300 K, there was a more intense decrease in the intensity of diffraction peaks, as well as the emergence of new peaks on diffractograms. The change of distorting factors in the structure was due to the accumulation of helium with the subsequent formation of helium inclusions, leading to deformation and degradation of the crystal lattice. When fluence exposure 5.0 × 10^17^ ion/cm^2^ was observed the formation of broadening of diffraction peaks with the formation of the amorphous halo took place, the presence of which is associated with partial destruction of the crystal and the chemical bonds in the lattice due to the accumulation of helium in the structure. Furthermore, for samples irradiated at a temperature of 1000 K, the dynamics of changes in diffraction maxima were less intense. The greatest change was associated with a decrease in intensities of the diffraction maxima, as well as a shift of the maxima to the region of small angles. In this case, the asymmetries of peaks were practically not observed.

One of the indicators of the influence of distortions and deformations of the structure was the estimate of the half-widths of the diffraction maxima using the Williamson–Hall method [40,41,42,43,44]. Figure 7b,d shows the Williamson–Hall construction for samples under study, as well as the magnitude of stresses in samples. According to the Williamson–Hall data, it can be seen that in the case of samples irradiated at high temperatures, the magnitude of the distortions was less, which was due to the partial annealing of defects and dislocations in the structure during thermal heating. In the case of samples irradiated at a temperature of 300 K, not only disordering regions and helium inclusions, but also vacancy defects resulting from elastic and inelastic collisions of ions with lattice atoms contributed to the distortion and deformation of the structure. Figure 9 shows the dynamics of changes in main crystallographic characteristics of distortions and deformations of the structure, such as the magnitude of RMS displacement of atoms from lattice sites, macrostresses, changes in density of surface layer, as well as the concentration of defect inclusions in the structure as a result of irradiation.

The magnitude of RMS displacements of atoms from lattice sites characterizes static distortions of the crystal structure associated with the presence of dislocation defects, interstitial atoms, and thermal vibrations of atoms of the lattice sites in the structure of dislocation defects. As can be seen from the presented data, a slight increase in the standard deviation of atoms from lattice sites for samples irradiated at a temperature of 1000 K, in contrast to samples irradiated at a temperature of 300 K, was due to an increase in thermal vibrations of atoms and the formation of helium inclusions in the structure. In this case, a decrease in the density of dislocations in the structure during high-temperature irradiation led to a decrease in macrostrains in the structure, as well as a smaller decrease in density of the surface layer. The decrease in density was due both to the appearance of distortions and areas of disorder in the structure, and to an increase in the concentration of defects in the surface layer of ceramics. In this case, thermal action led to partial annihilation of defects and vacancies in the structure, which led to less radiation damage to the structure and a decrease in the magnitude of helium swelling. However, irradiation at high temperatures led to an increase in thickness of the damaged layer due to an increase in the penetrating ability of helium into the material. The presence of a polycrystalline structure led to a slight reorientation of the texture at low concentrations of implanted helium in the structure at room temperature, while under irradiation at high temperatures, the presence of different texture directions led to an increase in the penetrating ability of helium.

## 4. Conclusions

The paper presents the result of the study of radiation helium swelling of nitride ceramics irradiated at temperatures of 300 and 1000 K. For samples irradiated at 1000 K, there was a smaller change in the hardness of the surface layer, however, a significant increase in the depth of the defect region exceeding 350–400 nm for maximum radiation dose. This increase in the depth of the defect zone might be due to the high mobility of helium ions in the structure, as well as accelerated migration into the material. It was established that at high irradiation temperatures, an increase in the thickness of the damaged layer was observed, but the degree of damage was much less than in the case of irradiation at room temperatures. The change in the density and deformation of the crystal structure was due to the accumulation of helium in the structure of the surface layer of ceramics with the subsequent formation of helium inclusions, the average size of which ranged from 200 to 500 nm. In the case of irradiation at a temperature of 300 K with an irradiation fluence of 5.0 × 10^17^ ion/cm^2^, two generations of helium inclusions were observed, which indicated a partial degradation of the surface layer due to flecking processes.

## Figures and Tables

**Figure 1 materials-12-02415-f001:**
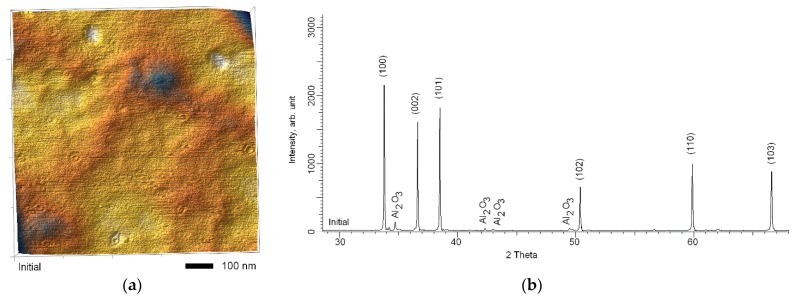
(**a**) AFM image of the surface of initial sample; (**b**) X-ray diffractogram of initial sample; (**c**) SEM image of the surface of initial sample.

**Figure 2 materials-12-02415-f002:**
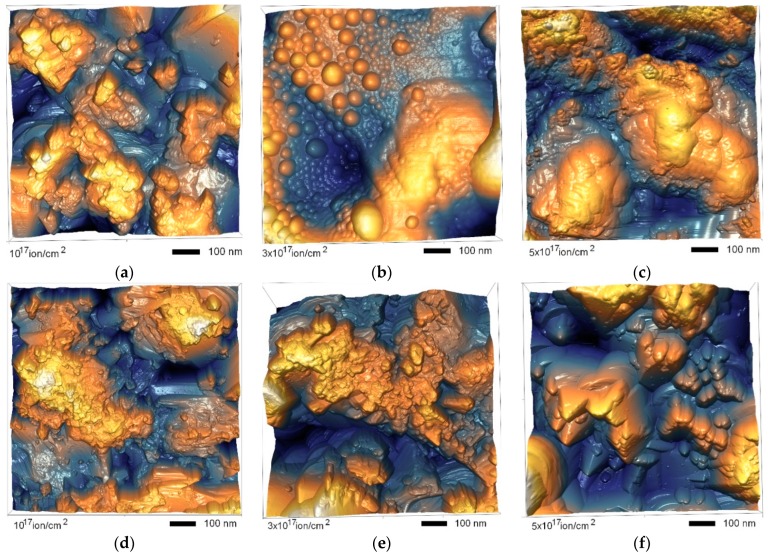
AFM images of studied ceramics after irradiation. (**a**) 10^17^ ion/cm^2^, 300 K; (**b**) 3 × 10^17^ ion/cm^2^, 300 K; (**c**) 5 × 10^17^ ion/cm^2^, 300 K; (**d**) 10^17^ ion/cm^2^, 1000 K; (**e**) 3 × 10^17^ ion/cm^2^, 1000 K; (**f**) 5 × 10^17^ ion/cm^2^, 1000 K.

**Figure 3 materials-12-02415-f003:**
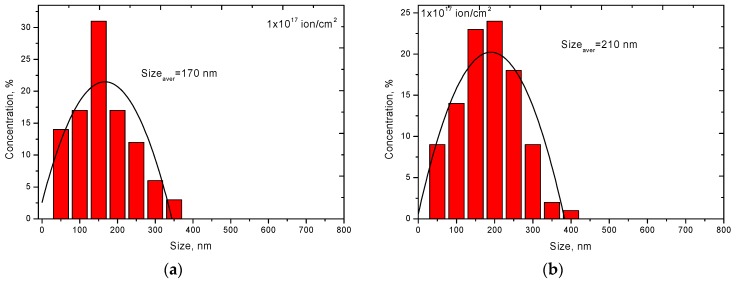
Dynamics of changes in the size of helium inclusions in the surface layer. (**a**) 10^17^ ion/cm^2^, 300 K; (**b**) 3 × 10^17^ ion/cm^2^, 300 K; (**c**) 5 × 10^17^ ion/cm^2^, 300 K; (**d**) 10^17^ ion/cm^2^, 1000 K; (**e**) 3 × 10^17^ ion/cm^2^, 1000 K; (**f**) 5 × 10^17^ ion/cm^2^, 1000 K.

**Figure 4 materials-12-02415-f004:**
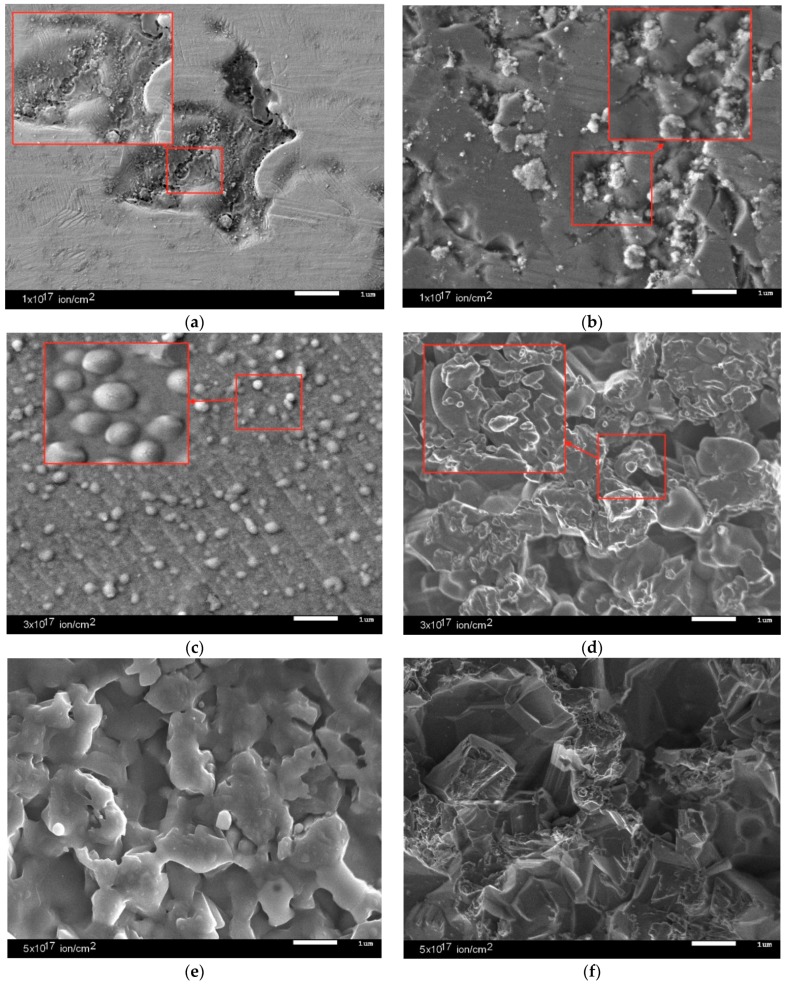
SEM images of samples surface after irradiation. (**a**) 10^17^ ion/cm^2^, 300 K; (**b**) 3 × 10^17^ ion/cm^2^, 300 K; (**c**) 5 × 10^17^ ion/cm^2^, 300 K; (**d**) 10^17^ ion/cm^2^, 1000 K; (**e**) 3 × 10^17^ ion/cm^2^, 1000 K; (**f**) 5 × 10^17^ ion/cm^2^, 1000 K.

**Figure 5 materials-12-02415-f005:**
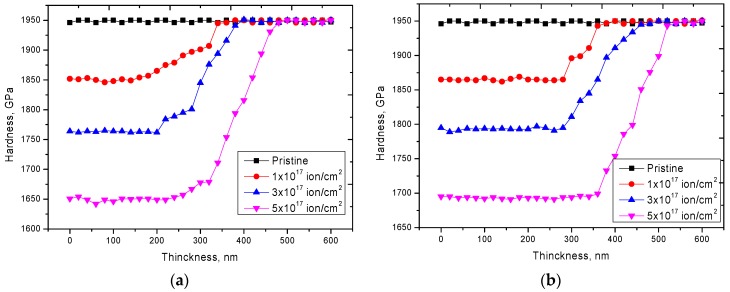
Dynamics of microhardness changes by depth depending on the irradiation conditions. (**a**) 300 K, (**b**) 1000 K.

**Figure 6 materials-12-02415-f006:**
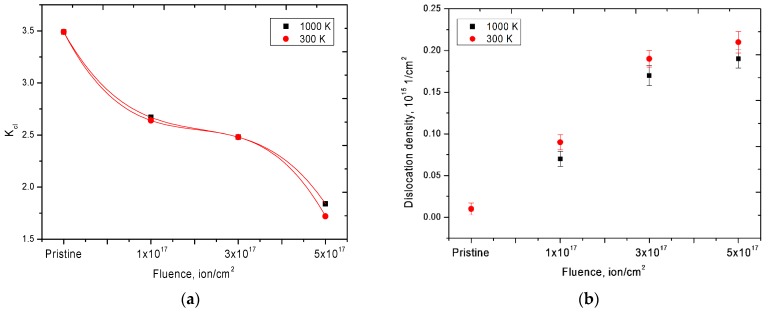
(**a**) Dynamics of change in the Griffith criterion; (**b**) graph of changes in the density of dislocations as a result of irradiation.

**Figure 7 materials-12-02415-f007:**
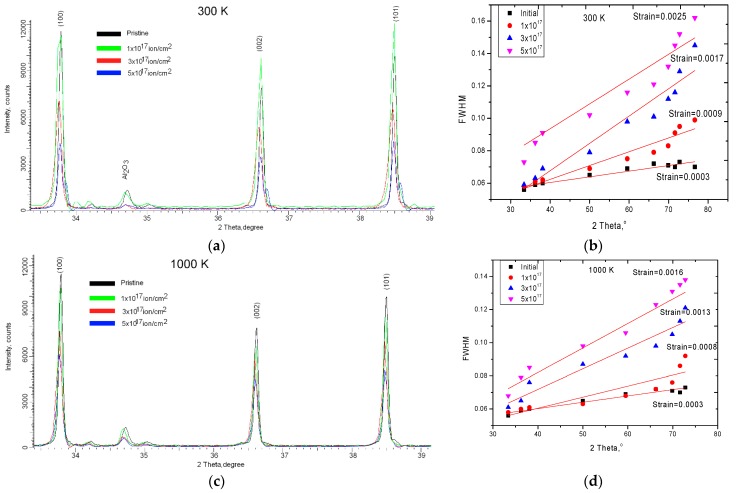
(**a**,**c**) X-ray diffraction patterns of studied samples; (**b**,**d**) construction of Williamson–Hall.

**Figure 8 materials-12-02415-f008:**
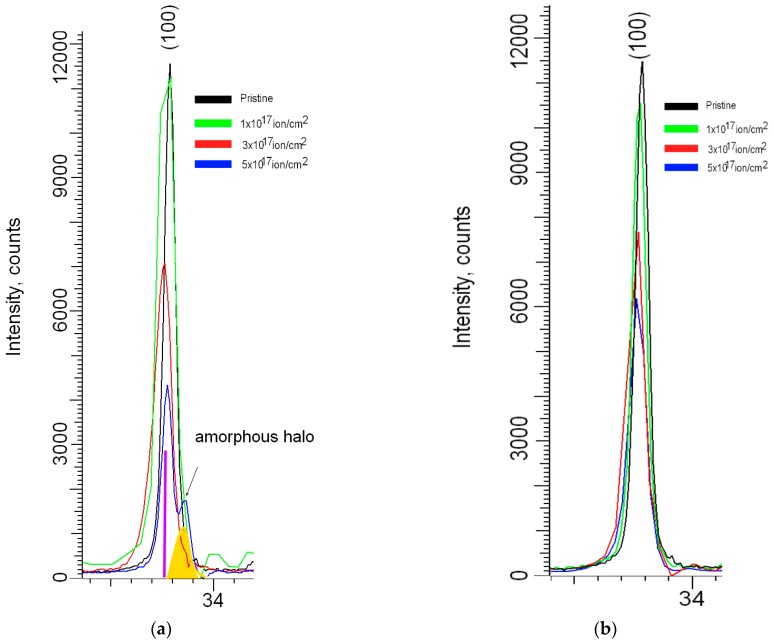
Dynamics of change of the main diffraction peak (100) depending on the irradiation fluence and temperature (**a**) 300 K; (**b**) 1000 K.

**Figure 9 materials-12-02415-f009:**
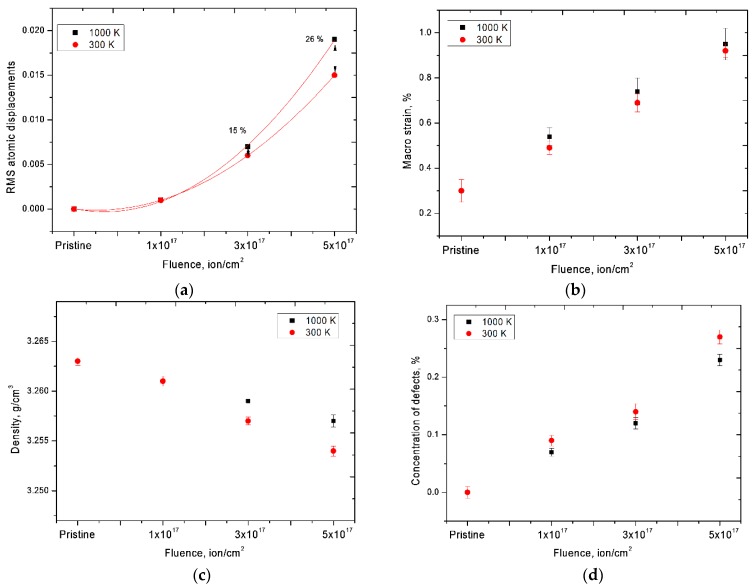
(**a**) Dynamics of change in the magnitude of atomic displacements from the lattice sites; (**b**) dynamics of changes in macrostress due to irradiation; (**c**) graph of the change in the density of ceramics as a result of irradiation; (**d**) dynamics of changes in the concentration of defects in the structure of ceramics.

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
