# Peer review of "Study of Helium Swelling in Nitride Ceramics at Different Irradiation Temperatures"

_materials, 2019, doi:10.3390/ma12152415_

Round 1

Reviewer 1 Report

The subject of the article is quite interesting and prospective. The text itself looks rather acceptable from the viewpoint of language, originality and contribution to the field of research. However, there are few drawbacks preventing immediate acceptance the manuscript in a present form. Below are some comments and suggestions.

1. Introduction.  The sentence “As a rule, ceramics based on aluminum nitride is obtained by sintering oxide

powders in a nitrogen-containing medium at high temperatures and pressures” seems to sound too categorically. There are several techniques to fabricate AlN (see, e.g., F. Jean-Marie Haussonne (1995) Review of the Synthesis Methods for AIN, Materials and Manufacturing Processes, 10:4, 717-755, DOI: 10.1080/10426919508935062) from alumina, so impurities could be consisted not only of alumina particles, but also of aluminium carbide (if carbonitridization was applied) , aluminium oxonitrides. Cited refs. 36 and 37 do not deal, in fact, with the issue above, but rather with thin film sputtering (ref.36) and metallization of ceramics (ref.37). Perhaps, these references should be replaced with reasonable ones, and AlN sintering routes have to be touched more carefully.

2. Materials and Methods section looks weird. Firstly, the origin of AlN sample is unknown. If there is a supplier, it should be specified. If somebody donated the samples, this person should be acknowledged; and better to describe how the samples were made. Anyway, this info on sample origin  is important to learn what we can expect in sense of impurities.  Secondly, there is no any description of devices used for irradiation and analysis (diffractometer, scanning microscope, AFmicroscope). Citing in this sense previous works of the authors seems not to be acceptable. Finally, in Fig.1 (XRD spectrum), some weak peaks are not marked. Do they also relate to alumina? This issue should be clarified.

3. The section “Results”. Perhaps, should be “results and discussion” since any discussion of the results is imperative. Fi.6, p.7, a) what authors mean on Griffith criterion (note the wrong writing “Griffits” when labeling y-axis) ? Usually, it is a condition for crack growth balancing strain energy and surface energy around the crack, not a single number. How, then, a condition can be placed as an axis label? May be, some dimension(less) ratio constructed from values entering this criterion is underlined? b) Dislocation density in ion/cm2 (see y-axis in b-picture) is erratum. Perhaps, should be cm-2, but in this way, dislocation density of about 1014 cm-2 looks extremely high for such a brittle body as AlN (compare with 1012 cm-2 being typical for heavily worked metals). Please, a) give a formula for Griffith criterion (or whatever it means) in the section “Materials and Methods”. Line 163-165 are not satisfactory: if one tells on crack resistance, relevant description (perhaps, in fracture toughness KIc) should be done. b) Give (in “Materials and Methods” or anywhere) the procedure of assessment of dislocation density.

Lines 181-184. Not only (101) line but all in Fig.7 c decreases their (integral) intensities with fluence increasing. It is not because of texture, in that case, only lines belonging to a certain zone will change their intensities. This is also not due to microstraining of a sample since in that case (typical straining field varies as 1/r like for dislocations) one will have line broadening. However, in the case point defects with strain field varying like 1/r3, it will contribute to integral intensity decreasing.

Lines 190-193. “The presence of additional maxima and broadening of the diffraction maxima with an irradiation fluence of 5.0x1017 ion/cm2 is observed due to the occurrence in the structure of a large number of dislocation defects and disordered regions resulting from helium swelling of the surface layer.” This looks not very plausible. Dislocations can contribute to line broadening, point defects to intensity decreasing, but why dislocations can generate any periodical lattice (the reason why additional reflexes can occur)? In general , the section of lines 181-209 (as well as 217-233) looks very dim and should be improved.

Conclusion. The manuscript under peer review is interesting and, surely, devote  publication. At the same time the text work  to un-check referee’s comments and notes is necessary.  Accepting of the manuscript is possible after re-reviewing of the new version.

Author Response

 As a rule, aluminum nitride-based ceramics are obtained by sintering oxide powders in a nitrogen-or carbon-containing medium at high temperatures and pressures. At the same time, the sintering temperature ranges from 660 to 2000 K, the choice of which is due to the chemical reactions of synthesis [36,37]. So in the works [36,37] a detailed description of methods for producing aluminum nitride by various methods obtained in different laboratory conditions is presented. In most cases, the sintering of nitride ceramics is accompanied by the release of gases CO, N2, H2.  Also, the reaction products in the synthesis process, depending on the gas medium, can be oxide, oxonitride, carbide inclusions and gas-filled pores. In most cases, industrial production of aluminum nitride is carried out using the method of carbothermic reduction of powders Al2O3 and C(s) in the medium N2(g) at a temperature of about 2000 K at a pressure p=0.12-0.26 atm. As a result of annealing, AlN-type ceramics and CO(g) gas are obtained, with a small part of Al2O3 being equally likely to dissolve in the ceramic structure.   At the same time, the appearance of impurity inclusions can lead to additional distortions and deformations of the crystal structure, as well as changes in the phase composition of ceramics.

The original ceramic samples were purchased from Mingrui Ceramic Tech. Co. Ltd. (Guangdong,China) specializing in the production of various ceramics for industrial applications and research.

The study of changes in structural characteristics and morphology of samples was carried out using atomic force microscopy (AIST-NT SPM microscope, AIST-NT Inc, Novato, CA 94949, USA, Scan mode is the following: AC-Mode (Non-Contact Mode), scan rate/scan frequency is 0.4 Hz, resolution XY 700x700, scan size XY 15x15 um, Z is automatic), X-ray diffraction (XRD D8 ADVANCE ECO diffractometer (Bruker, Karlsruhe, Germany) using CuKα radiation, identify the phases and study the crystal structure, the software BrukerAXSDIFFRAC.EVAv.4.2 and the international ICDD PDF-2 database), and scanning electron microscopy (SEM, Hitachi TM3030 (Hitachi Ltd., Chiyoda, Tokyo, Japan). A detailed description of the methods and instrumentation base is presented in our works [38-44].

Helium swelling was simulated by irradiating nitride ceramics with low-energy He2+ ions with an energy of 40 keV at temperatures of 300 and 1000 K with a fluence of 1x1017 - 5x1017 ions/cm2. Irradiation was carried out at the DC-60 heavy ion accelerator (Nursultan,Kazakhstan). The ion path lengths were calculated using the Kinchin-Pease model using the SRIM Pro 2013 program and 240±10 nm, the radial deviation is 60±5 nm, the energy loss of ions on electrons is dE/dxelec = 0.184 keV/nm, the energy losses on nuclei are dE/dxnuclear = 0.07 keV/nm [45].

As a rule, the dislocation density in ceramic materials in the initial state is 1014-1016 cm-2, such a value of dislocation density is due to the small size of grains or crystallites, as well as the presence of various impurities affecting the dislocation density and its distribution near the grain boundaries. At the same time, an increase in the dislocation density in the structure above 1017 cm-2 leads to a sharp degradation of ceramics, due to the partial destruction of crystal and chemical bonds, as well as the fragmentation of crystallites as a result of external influences. According to the data obtained, the dislocation density in the structure of ceramics does not exceed 1015 cm-2, which confirms the data on the high strength of ceramics after irradiation.

As can be seen from the presented data, for the samples irradiated at a temperature of 300 K at a fluence of irradiation of 1.0x1017 ion/cm2 there is a slight change in the intensity of the diffraction maxima, as well as the broadening of the diffraction lines. A slight displacement of the maxima in the region of small angles indicates a change in interplanar distances and crystal lattice parameters as a result of irradiation. An increase in the irradiation fluence leads to a decrease in the intensity and an increase in the asymmetry of the diffraction maxima, which indicates an increase in distortion and deformation in the structure as a result of irradiation. The formation of helium inclusions in the structure leads to a partial disorientation of texture directions, as well as a change in the size of crystallites, as evidenced by the change in dislocation density. The broadening of the diffraction peaks can be caused both by the size effect and by distorting factors that arise as a result of irradiation.

The change in distorting factors in the structure is due to the accumulation of helium with the subsequent formation of helium inclusions, leading to the deformation and degradation of the crystal lattice. Also, for samples irradiated at high temperatures with a maximum irradiation fluence, no additional maxima appear. Figure 8a-b shows the comparative dynamics of the main diffraction maximum (100) depending on the fluence and the irradiation temperature. As can be seen from the presented data at the irradiation temperature of 300 K there is a more intense decrease in the intensity of the diffraction peaks, as well as the emergence of new peaks on diffractograms. The change of distorting factors in the structure is due to the accumulation of helium with the subsequent formation of helium inclusions, leading to deformation and degradation of the crystal lattice. When fluence exposure 5.0х1017 ion/cm2 is observed the formation of the broadening of the diffraction peaks with the formation of the amorphous halo, the presence of which is associated with partial destruction of the crystal and the chemical bonds in the lattice due to the accumulation of helium in the structure. Also, for samples irradiated at a temperature of 1000 K, the dynamics of the diffraction maxima is less intense. The greatest change is due to the decrease in the intensity of the diffraction maxima, as well as the displacement of the maxima in the region of small angles. In this case, the asymmetry of the peaks is practically not observed.

Reviewer 2 Report

This paper reports the effect of the He-ion irradiation fluence and temperature on the microstructures, defect structures and mechanical properties of AlN ceramics. This work provides some meaningful information. However, the reviewer feels that the English language and style in the paper seem not to reach the required level of Materials. Although the past tense is used in section 2 about Materials and Methods, the present tense is used in section 3 about Results. The past tense is generally used for the results obtained. From this point, the English language and style are not appropriate as scientific writing. But the contents in this manuscript are interesting. The reviewer suggests the authors modify the English grammatical structures for the next submission.  

Some minor issues:

1. (Page 1, Line 38-39)(Page3, Line 90-92): Is this English correct?

"and in the case of surface layer contact the with the coolant "

"an increase in the dose of radiation above 1.0x1017 ion/cm2 at an irradiation temperature of 300 K in the surface layer is observed the formation of spherical inclusions whose average size varies from 300 to 400 nm."

2. (Page 8, Line 181-184): To discuss the effect of the irradiation fluence on the microstructure of the sample, an SEM image of the surface of the sample before irradiation should be necessary. The authors stated a change in the orientation of crystallites and the direction of the textures by irradiation with 1*107 ions cm-2 and at 300 K; however, from the SEM image of the sample irradiated in Figure 4, the morphology of the sample seems not significantly change. 

3. (Page 8, Line 190): What does "the presence of additional maxima" mean in the XRD patterns of Figure 7? 

4. It is recommended to add error bars on experimental data points in Figures 6 and 8 to mention the effect of recombination of defects on the dislocation density and strain. The accuracy of the experimental data should be addressed in the paper.

Author Response

1. 

Corrected. English fixed

"and in the case of surface layer contact with the coolant "

"an increase in the dose of radiation above 1.0x1017 ion/cm2 at an irradiation temperature of 300 K in the surface layer the formation of spherical inclusions whose average size varies from 300 to 400 nm is observed."

2. 

Figure 1. a) AFM image of the surface of initial sample; b) X-ray diffractogram of initial sample; c) SEM – image of the surface of initial sample

3. Figure 8a-b shows the comparative dynamics of the main diffraction maximum (100) depending on the fluence and the irradiation temperature. As can be seen from the presented data at the irradiation temperature of 300 K there is a more intense decrease in the intensity of the diffraction peaks, as well as the emergence of new peaks on diffractograms. The change of distorting factors in the structure is due to the accumulation of helium with the subsequent formation of helium inclusions, leading to deformation and degradation of the crystal lattice. When fluence exposure 5.0х1017 ion/cm2 is observed the formation of the broadening of the diffraction peaks with the formation of the amorphous halo, the presence of which is associated with partial destruction of the crystal and the chemical bonds in the lattice due to the accumulation of helium in the structure.

4. Measurement errors are added; methods of calculations are presented.

Round 2

Reviewer 2 Report

The revisions are satisfactory and thus I feel that your manuscript is worth publishing. But, as previously mentioned in the 1st review, the present tense for results obtained is inappropriate for scientific writing. I'm wondering why the authors consistently use the present tense for the results, even though the past tense is used in the Materials and Methods. It is reasonable to use the present tense for discussions; however, as far as I know, it is not common to use the present tense for the results. See the below links to understand why the past tense should be used for results obtained.

1. https://services.unimelb.edu.au/__data/assets/pdf_file/0009/471294/Using_tenses_in_scientific_writing_Update_051112.pdf

2. https://www.nature.com/scitable/topicpage/effective-writing-13815989

3. https://www.editage.com/insights/which-tense-should-be-used-in-the-results-and-discussion-section-of-a-paper

Author Response

Corrections made.